# Fusion-Augmented Large Language Models: Boosting Diagnostic Trustworthiness via Model Consensus

Md Kamrul Siam*, Md Jobair Hossain Faruk*, Jerry Q. Cheng*, Huanying Gu*

*Department of Computer Science, New York Institute of Technology, USA
{mhossa37, ksiam01, jcheng18, hgu03}@nyit.edu

*Abstract*—This study presents a novel multi-model fusion framework leveraging two state-of-the-art large language models (LLMs), ChatGPT and Claude, to enhance the reliability of chest X-ray interpretation on the CheXpert dataset. From the full CheXpert corpus of 224,316 chest radiographs, we randomly selected 234 radiologist-annotated studies to evaluate unimodal performance using image-only prompts. In this setting, ChatGPT and Claude achieved diagnostic accuracies of 62.8% and 76.9%, respectively. A similarity-based consensus approach, using a 95% output similarity threshold, improved accuracy to 77.6%. To assess the impact of multimodal inputs, we then generated synthetic clinical notes following the MIMIC-CXR template and evaluated a separate subset of 50 randomly selected cases paired with both images and synthetic text. On this multimodal cohort, performance improved to 84% for ChatGPT and 76% for Claude, while consensus accuracy reached 91.3%. Across both experimental conditions, agreement-based fusion consistently outperformed individual models. These findings highlight the utility of integrating complementary modalities and using output-level consensus to improve the trustworthiness and clinical utility of AI-assisted radiological diagnosis, offering a practical path to reduce diagnostic errors with minimal computational overhead.

*Index Terms*—Large Language Models, Multimodal Fusion, LangChain, Trustworthy AI, MIMIC CXR, AI Healthcare

## I. INTRODUCTION

LLMs have demonstrated strong capabilities in natural language understanding, contextual reasoning, and zero-shot inference, spurring increasing interest in their application to clinical decision support [1]. In radiology, diagnostic reasoning relies not only on imaging but also on nuanced clinical context such as patient history, physical findings, and prior studies [2]. The ability of LLMs to integrate diverse inputs and generate structured outputs offers opportunities to support radiologic workflows in tasks such as report generation, abnormality detection, and diagnostic triage [3], [4]. However, recent studies show that even state-of-the-art LLMs can produce hallucinations or overlook subtle findings, limiting their trustworthiness in high-stakes domains like radiology [5].

Despite recent advances, LLM performance in medical imaging remains variable. Prior evaluations have shown that even advanced models often struggle with ambiguous findings or complex radiologic nuances, raising safety concerns in real-world use [6], [7]. Moreover, most LLMs operate primarily on textual inputs, lacking native support for multimodal reasoning. As a result, integrating rich diagnostic signals-especially from both text and image-is critical to enabling trustworthy, comprehensive decision support in radiology.

To address this, we propose a model-agnostic framework that orchestrates multiple LLMs through prompt-based co-ordination and semantic agreement, without requiring any model fine-tuning or retraining. Our approach aims to improve diagnostic reliability by leveraging output-level consensus and incorporating multimodal inputs, such as synthetic clinical notes paired with medical images.

As an ideal testbed for evaluating AI-assisted image interpretation, the CheXpert dataset [8], with over 200,000 labeled chest X-rays across 14 pathology categories, has become a standard benchmark for model validations. Recent studies have applied LLMs to this dataset, with varied diagnostic performance has, particularly in cases involving uncertainty or subtle findings [9]. One established strategy to improve predictive reliability in machine learning is ensemble learning, combining multiple models to leverage their individual strengths and reduce susceptibility to errors [10]. While traditional ensemble learning methods often use techniques like majority voting or model averaging, applying these principles to LLMs presents unique challenges. Because LLMs typically return unstructured text responses, they are accessible only through APIs with limited transparency, and require alignment across multiple modalities. In clinical applications, ensemble outputs must also be interpretable and auditable, supporting clinicians in understanding the basis of automated recommendations.

To address these challenges, this paper introduces a consensus-based fusion framework that coordinates multiple LLMs using LangChain [11], an open-source framework designed for building applications with large language models through modular prompt chaining and tool integration. Our system performs parallel inference across models, compares outputs using semantic and clinical alignment metrics, and flags high-confidence predictions based on inter-model agreement. Designed to mimic the clinical practice of seeking a second opinion, the framework emphasizes explainability and supports both unimodal and multimodal inputs for diagnostic tasks. The primary contributions of this work are as follows:

- We propose a modular, model-agnostic framework for LLM-based diagnostic interpretation that supports both

The work was partially supported by the National Science Foundation Grants CNS-2120350 and III-2311598

unimodal and multimodal inputs.
- We design an interpretable consensus metric that leverages semantic similarity and label concordance to identify high-confidence predictions.
- We validate the framework using the CheXpert dataset and show that model agreement improves diagnostic reliability beyond individual models.

The remainder of this paper is organized as follows. Section II reviews relevant literature on LLM applications in medical imaging and ensemble methods. Section III describes the methodologies including details our dataset, multimodal data synthesis, and the LangChain orchestration framework. Section IV presents experimental results and analysis of model agreement and consensus accuracy. Section V discusses clinical implications, limitations, and avenues for future work. Finally, Section VI concludes the paper.

## II. RELATED WORK

### A. LLMs in Medical Decision-Making

LLMs like GPT-4 and Claude have shown strong performance in clinical tasks such as question answering and note generation [12], [13]. In radiology, studies increasingly evaluate LLMs for interpreting imaging findings when paired with textual descriptions or integrated into multimodal frameworks [14]–[16]. While these models can recognize common pathologies, they can, in some instances, hallucinate findings or miss subtle abnormalities, especially with limited context [17]. This observation is a key motivator for our work, which seeks to mitigate such potential failures through a consensus mechanism. Significant output variability across LLMs highlights the need for verification mechanisms. Divergent responses from models like GPT-4 and Claude 3 suggest that cross-validation can improve reliability, motivating our dual-LLM consensus approach. Recent efforts in medical-domain LLMs (e.g., MedLM [18], BioGPT [19]) and vision-language models (e.g., GPT-4V [20]) offer promising directions. However, these models still fall short of radiologist-level performance, particularly on complex imaging tasks [21], [22].

Our work builds on these insights, treating LLMs as collaborative advisors and fusing their outputs to improve trust and reduce diagnostic errors. By doing so, we aim to bridge the gap between the current capabilities of LLMs and the high reliability required for clinical decision-making.

### B. Ensemble Methods in AI

Ensemble learning enhances model robustness by combining diverse predictors using techniques like majority voting or bagging [23]. In medical imaging, ensembles are common in benchmarks such as CheXpert [8]. Numerous studies have demonstrated the value of ensemble approaches in medical imaging for tasks ranging from segmentation to classification [23], [24].However, LLM ensembles pose new challenges due to unstructured, free-text outputs.

Existing work has explored intra-model consistency [25], but cross-model ensembles remain limited. We extend ensemble principles to LLMs by applying semantic similarity

scoring (e.g., BERTScore [26]) across independently generated outputs. This enables interpretable, threshold-based consensus without retraining. To ensure AI safety, establishing agreement between independent agents can prevent diagnostic errors [24].

### C. Multimodal Fusion in Medical AI

Clinical diagnosis inherently relies on multimodal data; imaging, notes, labs, etc. [27]. For chest radiography, combining image interpretation with textual context improves accuracy. The CheXpert dataset exemplifies this by linking images with report-derived labels [8]. CNNs and transformers for image and text have improved complex pathology detection in recent models. [28].

Our approach encodes chest X-rays into base64 format and provides them as string to vision-capable LLMs or text-only models through staged prompts, in addition to synthetic clinical notes. This design is modeled after radiologic procedures, which involve the simultaneous evaluation of the clinical context and the image. Despite the fact that vision-language models possess inherent multimodal capabilities, their efficacy on clinical imaging is still restricted [29]. This study also offers a pragmatic fusion strategy that is compatible with existing LLM infrastructures, allowing for diagnostic reasoning without the need for model fine-tuning or alterations to the architecture.

## III. METHODOLOGY

### A. Dataset and Preprocessing

The dataset used in this study is the publicly available CheXpert dataset, developed by the Stanford Machine Learning Group[1]. It consists of 224,316 chest radiographs from 65,240 patients and includes posteroanterior (PA), anteroposterior (AP), and lateral views. To construct our evaluation sets, we randomly selected 234 studies from this pool for the unimodal task. The random sampling was performed to ensure the selected subset provides a representative, unbiased sample of the various pathologies and patient demographics present in the full dataset. A further random subset of 50 cases was chosen for the multimodal analysis.

In PA imaging, the X-ray beam passes from the patient's back to front with the patient standing upright. This projection minimizes cardiac magnification and is the gold standard for diagnostic chest X-rays. In contrast, AP images, typically acquired from bedridden or ICU patients, are more prone to distortion, including heart magnification and overlapping anatomical structures. Lateral views complement PA/AP images, offering a side view of the chest, which aids in diagnosing conditions like pleural effusion, pneumonia, and lung lesions. Each image in the dataset is labeled for the presence or absence of 14 thoracic conditions: *enlarged cardiomediastinum, cardiomegaly, lung opacity, lung lesion, edema, consolidation, pneumonia, atelectasis, pneumothorax, pleural effusion, pleural other, fracture, support devices*, and *no finding*. Labels were generated by a rule-based NLP labeler

---

[1]https://stanfordmlgroup.github.io/competitions/chexpert/

and were reviewed by radiologists. The label "no finding" is assigned when none of the other 13 abnormalities are present, ensuring a clean set of normal radiographs for model evaluation.

**Image Preprocessing and Encoding:**

To prepare the images for multimodal LLM-based analysis, a standardized preprocessing and encoding pipeline was applied. If an image was in DICOM format (.dcm), it was loaded using the `pydicom` library, normalized to 8-bit grayscale, and converted to the PIL image format. Non-DICOM images (e.g., JPEG/PNG) were processed using the Python Imaging Library (PIL) and converted to RGB color format for consistency. All images were saved as JPEGs in a structured directory and subsequently encoded into base64 strings to facilitate transmission in JSON-compatible format, along with the associated MIME type. These base64-encoded images were passed into a LangGraph workflow, where the image path was extracted from the current system state, and the image was encoded with its MIME type for subsequent processing.

**Model Input Formatting and Execution:**

Each base64-encoded image, together with its MIME type and a short "question" string, was wrapped into a single JSON payload and sent to both OpenAI's GPT-4 and Anthropic's Claude 3. To standardize both models' behavior, we employed the following prompt template:

```
Assume you are a radiology assistant. Based
on given X-ray(s) and its clinical context,
identify any possible or tentative diseases
out of the following list (or none):
Enlarged Cardiomediastinum, Cardiomegaly,
Lung Opacity, Lung Lesion, Edema,
Consolidation, Pneumonia, Atelectasis,
Pneumothorax, Pleural Effusion, Pleural
Other, Fracture, Support Devices. If you
identify any, output a single digit "1,"
otherwise output "0."
Note: For research use only. Do not use for
clinical decisions.

Context: {context}
Question: {question}

Respond in this format:
POSSIBLE DIAGNOSES:
<either 1 or 0>
```

This forces a clean, one-character response per finding, which we parse into our 14-dimensional prediction vector. Both LLMs received the exact same prompt and image data, ensuring a fair side-by-side comparison.

**Consensus Fusion:** We then compared the two model outputs using BERTScore. While a simple exact match is suitable for binary outputs, we chose BERTScore for its ability to capture semantic similarity, making our framework more adaptable for future extensions with more complex, free-text outputs. For this study, we selected a stringent 95% threshold for agreement to ensure high confidence in the consensus results. Cases with score $\geq 95\%$ were accepted as consensus; any lower score triggered a manual review flag.

**Experimental Conditions:**

In this study, medical images from the CheXpert dataset were used. Images were selected randomly from the dataset to minimize potential biases, ensuring that factors such as age, race, and gender were not considered during the selection process.

1) **Unimodal Setup:** A total of 234 curated chest X-rays, including both frontal (PA/AP) and lateral views, were analyzed using image-only input.
2) **Multimodal Setup:** A random subset of 50 cases, selected from the 234 curated chest X-rays, was analyzed using both the encoded image and the corresponding synthetic clinical notes (described in Section III-B).

### B. Synthetic Clinical Note Generation

Access to authentic radiology reports is severely constrained by privacy regulations, de-identification overhead, and the need for expert review [30]. One of the key challenges we encountered when working with both public and private datasets-including CheXpert-was the lack of paired clinical notes alongside imaging data. To address this limitation and explore whether even minimal, templated clinical guidance can enhance multimodal performance with large language models (e.g., OpenAI's GPT, Anthropic's Claude), we generated synthetic clinical notes that encode only the presence or absence of each finding. This approach preserves patient privacy and scalability while providing richer context than image-only inputs.

While CheXpert includes binary expert labels for 14 thoracic conditions, it does not contain narrative reports [8]. To simulate a realistic diagnostic workflow, we constructed synthetic notes using a five-part template modeled on MIMIC-CXR-Examination, Indication, Technique, Findings, and Impression [31]. For each condition (e.g., Atelectasis, Cardiomegaly, Lung Opacity), we inserted predefined phrases in formal radiological language: positive statements for labels marked 1 and negated phrases for those marked 0.

To promote lexical variation while preserving semantic fidelity, we sampled from synonym pools and randomized sentence order. This yielded clinically coherent yet textually distinct notes that aligned strictly with the CheXpert label vectors. An excerpt from a real MIMIC-CXR note and its synthetic counterpart is shown in Figure 1, illustrating how our pipeline mirrors authentic report structure without exposing protected health information.

To give additional context on the underlying classification task, we reproduce the original CheXpert schematic from Irvin *et al.* in Figure 2, which visualizes the multi-label prediction challenge over frontal and lateral chest radiographs.

Each synthetic note was embedded using OpenAI's text-embedding-ada-002 model (1536-dim, via LangChain). We then computed cosine similarity between LLM-generated reports to assess semantic alignment, supporting fine-grained agreement scoring in the consensus fusion stage.

### C. Model Pipeline and Fusion Framework

Our diagnostic workflow is structured into three distinct stages as shown in Figure 3:

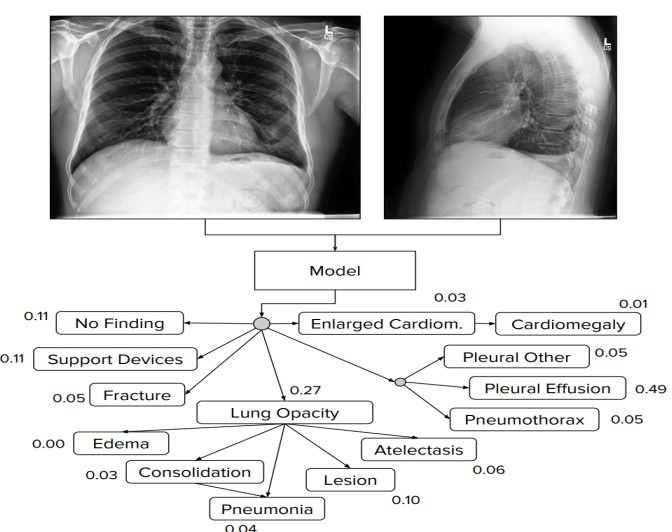

(a) Real MIMIC-CXR report excerpt     (b) Generated synthetic note

Fig. 1: Comparison between (a) a de-identified snippet from a MIMIC-CXR clinical report and (b) an example of synthetic note generated for our experiment.

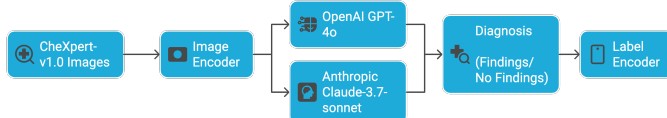

Fig. 2: CheXpert: predict the probability of 14 findings from the frontal and lateral chest radiographs (from Irvin *et al.* [8]).

Fig. 3: Proposed AI-Driven Medical Diagnosis Workflow

1) Embedding Extraction: Chest X-rays were converted into 512-dimensional embeddings using the CLIP (ViT-B/32) model. Synthetic clinical notes were embedded into 768-dimensional vectors via BioClinicalBERT. Image-text alignment was validated through cosine similarity checks:

$$\text{Cosine Similarity}(X, Y) = \frac{X \cdot Y}{|X||Y|}$$

2) Parallel LLM Inference: We employed two LLMs, Chat-GPT (GPT-4) and Claude (3.7-sonnet), which independently provided binary predictions (1 for present, 0 for absent) for each of the 14 diagnostic labels. A consistent and structured prompt template was employed for standardization.

3) Consensus Fusion: The semantic similarity of both models' outputs was assessed using BERTScore as:

$$\text{BERTScore}(P, R, F1) = \frac{2 \cdot P \cdot R}{P + R},$$

where $P$ and $R$ represent precision and recall scores, respectively. A threshold of 95% similarity was set to define consensus. Predictions exceeding this threshold were accepted as reliable consensus; otherwise, cases were flagged for manual review.

### D. Statistical Analysis

We used McNemar's test to statistically evaluate the significance of performance differences between individual models and the consensus. The test statistic is calculated as:

$$\chi^2 = \frac{(b - c)^2}{b + c},$$

where $b$ represents the number of cases correctly predicted by the individual model but incorrectly by the consensus, and $c$ represents the opposite.

### E. Unimodal Analysis

In the unimodal analysis, diagnostic predictions are generated exclusively from imaging data without auxiliary textual context. Each large language model (LLM), ChatGPT (GPT-4) and Claude (3.7-sonnet) produces independent binary predictions (presence or absence) for each of the 14 thoracic conditions from the chest X-ray embeddings. Individual model accuracy is calculated as:

$$\text{Accuracy} = \frac{\text{Number of Correct Predictions}}{\text{Total Number of Cases}}$$

Subsequently, model outputs are compared to identify consensus cases based on semantic similarity evaluated using BERTScore. Predictions from both models are considered to agree if the BERTScore exceeds the predefined similarity threshold of 95%. The proportion of cases where the models exhibit such agreement is calculated as:

$$\text{Agreement Rate (\%)} = \frac{\text{Number of Agreement Cases}}{\text{Total Number of Cases}} \times 100$$

Finally, predictions agreed upon by both models are treated as fused consensus results, with accuracy calculated similarly to individual models. These consensus outcomes undergo statistical validation using McNemar's test to determine whether consensus fusion yields statistically significant improvements over individual model performances. Detailed quantitative evaluation results are presented in the Results section.

### F. Multimodal Analysis

In multimodal analysis, each diagnostic prediction integrates information from two distinct sources: chest X-ray images and synthetic clinical notes. The synthetic notes are generated using standardized radiological terminology inspired by the MIMIC-CXR report template. The dual-input strategy uses visual and textual information to complement each other.

Initially, two embeddings are extracted, one for image denoted as $E_{\text{img}} \in \mathbb{R}^{512}$ via CLIP (ViT-B/32), and another for text denoted as $E_{\text{text}} \in \mathbb{R}^{768}$ via BioClinicalBERT. Concatenating these embeddings creates a multimodal representation:

$$E_{\text{multimodal}} = [E_{\text{img}}; E_{\text{text}}]$$

The combined embeddings are input into each LLM (ChatGPT and Claude) separately, producing binary predictions (presence 1, absence 0) for each of the 14 conditions.

Model accuracy in the multimodal context is calculated similarly as:

$$\text{Accuracy}_{\text{multimodal}} = \frac{\text{Correct Multimodal Predictions}}{\text{Total Multimodal Cases}}$$

Consensus predictions are determined based on semantic similarity using BERTScore, with the threshold set at 95% similarity between models' outputs. The multimodal consensus agreement rate is thus evaluated as:

$$\text{Multimodal Agreement (\%)} = \frac{\text{Agreement Cases}}{\text{Total Multimodal Cases}} \times 100$$

Statistical significance for improvements observed via multimodal fusion (consensus) compared to unimodal results is verified using McNemar's test, the details and results of which are presented in the Results section.

## IV. RESULTS

### A. Unimodal Performance: Image Only

In the image-only unimodal analysis (N=234), ChatGPT achieved an overall accuracy of 62.8% (147 correct cases), while Claude reached a higher correctness of 76.9% (180 correct cases). Table I demonstrates 77.6% prediction accuracy using semantic similarity-based consensus for unimodal.

TABLE I: Unimodal Diagnostic Correctness ($N = 234$)

| Metric | ChatGPT | Claude | Consensus |
|---|---|---|---|
| Correctness (%) | 62.8 | 76.9 | 77.6 |
| Count (correct/total cases) | 147/234 | 180/234 | 132/170 |

Detailed confusion matrices (Table II) reveal model-specific performance characteristics. ChatGPT had notably higher false-negative cases, indicating difficulty in reliably detecting positive abnormalities. Claude outperformed ChatGPT, showing fewer false negatives and improved classification accuracy.

TABLE II: Confusion Matrices for Unimodal ($N = 234$)

| | ChatGPT | | Claude | |
|---|---|---|---|---|
| True | 0 (Neg) | 1 (Pos) | 0 (Neg) | 1 (Pos) |
| 0 (Neg) | 132 | 64 | 159 | 37 |
| 1 (Pos) | 23 | 15 | 17 | 21 |

Classification metrics in Table III further illustrate Claude's stronger performance, achieving higher precision, recall, and F1-score compared to ChatGPT.

TABLE III: Classification Metrics for ChatGPT and Claude

| Model | Accuracy | Precision | Recall | F1-Score |
|---|---|---|---|---|
| ChatGPT | 0.628 | 0.74 (weighted) | 0.628 (weighted) | 0.67 (weighted) |
| Claude | 0.769 | 0.82 (weighted) | 0.769 (weighted) | 0.79 (weighted) |

Table IV illustrates these four rates for the image-only setting. Model agreement analysis indicates ChatGPT and Claude agreed in 170 out of 234 cases (72.6%), with consensus accuracy in agreed cases reaching 77.6%.

TABLE IV: Model Agreement for Unimodal ($N = 234$)

| Description | Count | Percentage | Notes |
|---|---|---|---|
| Total cases | 234 | 100% | – |
| Agreement cases (Chat-GPT = Claude) | 170 | 72.6% | – |
|   Correct predictions | 132 | 77.6% | Both models correct |
|   Incorrect predictions | 38 | 22.4% | Both models incorrect |
| Disagreement cases | 64 | 27.4% | Models diverged |

Similarly, Statistical significance for unimodal is evaluated by McNemar's test confirmed consensus-based fusion significantly improved accuracy compared to ChatGPT alone ($\chi^2 = 12.96, p < 0.001$). However, improvement over Claude was not statistically significant ($\chi^2 = 0.111, p = 0.74$).

### B. Multimodal Performance: Image + Clinical Notes

*1) Comparative Analysis on the 50-Case Subset:* To ensure a fair comparison with our multimodal results, we first evaluated the unimodal performance on the same subset of 50 cases. On this subset, GPT-4 achieved a unimodal accuracy of 70.0% (35/50), and Claude 3 achieved 74.0% (37/50). This provides a direct baseline to assess the impact of adding synthetic clinical notes. We evaluated multimodal inputs combining imaging data with synthetic clinical notes in a randomly selected subset of 50 cases. Multimodal integration significantly improved ChatGPT's accuracy to 84.0% (42 correct cases) compared to its unimodal baseline, while Claude achieved 76.0% accuracy (38 correct cases). Notably, consensus fusion with multimodal inputs enhanced diagnostic accuracy to 91.3% in cases when both models agreed (42 accurate out of 46 agreed cases).

*2) Multimodal performance:* : The confusion matrices in Table VI demonstrate that ChatGPT outperformed Claude in recognizing positive situations with more true positives, despite both models benefiting from multimodal inputs. The

TABLE V: Diagnostic Correctness ($N = 50$)

| Metric | ChatGPT | Claude | Consensus |
|---|---|---|---|
| Correctness (%) | 84.0 | 76.0 | 91.3 |
| Count (correct/total cases) | 42/50 | 38/50 | 42/46 |

classification metrics in Table VII show that both models outperformed unimodal analysis in multimodal inputs, with ChatGPT achieving superior recall.

TABLE VI: Confusion Matrices for Multimodal ($N = 50$)

| | ChatGPT | | Claude | |
|---|---|---|---|---|
| True | 0 (Neg) | 1 (Pos) | 0 (Neg) | 1 (Pos) |
| 0 (Neg) | 7 | 4 | 7 | 4 |
| 1 (Pos) | 5 | 34 | 8 | 31 |

TABLE VII: Classification Metrics for Multimodal

| Model | Accuracy | Precision (Weighted) | Recall (Weighted) | F1-Score (Weighted) |
|---|---|---|---|---|
| ChatGPT | 0.82 | 0.89 | 0.87 | 0.88 |
| Claude | 0.76 | 0.89 | 0.79 | 0.84 |

Table VIII highlights the model agreement analysis and increased model concordance with multimodal data. Agreement rose substantially to 92.0% (46 out of 50 cases), with a consensus accuracy of 91.3% among these agreed cases.

TABLE VIII: Model Agreement for Multimodal ($N = 50$)

| Description | Count | Percentage | Notes |
|---|---|---|---|
| Total cases | 50 | 100% | – |
| Agreement cases (Chat-GPT = Claude) | 46 | 92.0% | – |
| Correct predictions | 42 | 91.3% | Both models correct |
| Incorrect predictions | 4 | 8.7% | Both models incorrect |
| Disagreement cases | 4 | 8.0% | Models diverged |

Similarly, McNemar's test for multimodal settings indicate no statistically significant differences between individual model accuracies and multimodal consensus (ChatGPT vs. Consensus: $\chi^2 = 0, p = 1.00$; Claude vs. Consensus: $\chi^2 = 1.6, p \approx 0.20$). However, the consensus accuracy remained superior numerically.

### C. Summary of Error and Disagreement Analysis

A qualitative review of the disagreement cases (i.e., Lack of consensus) revealed that they often involved subtle findings or pathologies with ambiguous visual evidence, such as early-stage lung opacity or mild cardiomegaly. In these instances, one model would often correctly identify the finding while the other missed it. This highlights the value of a multi-model approach in capturing a wider range of potential diagnoses. When both models were incorrect, the errors were typically on challenging cases with multiple co-occurring findings.

## V. DISCUSSION

### A. Implications for Clinical AI Deployment

The findings from our study strongly support the principle of "wisdom of the ensemble" in clinical AI systems: combining multiple independent models yields more reliable and trustworthy diagnostic outcomes compared to relying on any single model. This concept mirrors established clinical workflows in radiology, where difficult cases are often reviewed by multiple radiologists or escalated for specialist opinion to minimize diagnostic errors. Our fusion framework acts as an automated mechanism to this process. When GPT-4 and Claude 3 agree on a diagnosis, it constitutes a robust validation signal and in our experiments, consensus outputs are correct most of the time (91.3% in multimodal dataset and 77.6% in unimodal dataset), a rate that suggests potential for meeting safety and efficacy thresholds suitable for assistive AI tools highlighting high-confidence findings.

Conversely, cases where the models disagree signal uncertainty and warrant caution. This disagreement serves as a valuable "fallback strategy," ensuring that ambiguous cases receive expert attention rather than being processed with lower confidence. In practical deployments, these cases can be prioritized for human review, thereby optimizing radiologist workload. Recent studies found a significant reduction in review time, error, and bias, demonstrating real-world efficiency gains from consensus-guided triage [32].

Furthermore, multi-model consensus enhances clinician trust in AI systems. Black-box AI models are often mistrusted due to unpredictable errors. Cross-validation with multiple models must concur before outputting a diagnosis, helps build confidence. In disagreements, radiologists may immediately identify the source of ambiguity and focus on confusing discoveries by transparently presenting both model outputs, turning AI from an opaque oracle to a collaborative advisor.

### B. Limitations and Future Work

Although our results have shown potential, the study has several limitations which define clear avenues for future work:

- Dataset Size and Composition: The multimodal experiments were performed on a small dataset of 50 cases with synthetic clinical notes. Future work will involve testing the framework on larger, more diverse datasets with authentic clinical notes to improve generalizability.
- Model Diversity: We evaluated only two proprietary LLMs. Our model-agnostic framework is designed for expansion, and future iterations will include open-source and domain-specific models to increase error diversity and robustness. This also addresses the challenge of proprietary models becoming outdated.
- Consensus Mechanism: We chose the 95% BERTScore criterion based on empirical data from our research. Out of 50 multimodal test samples, 46 exhibited complete agreement (BERTScore = 1.0) and four had near-perfect agreement ($BERTScore \approx 0.915$). The average is 95.75%, which led us to choose a 95% threshold as a reasonable midpoint for identifying high-confidence consensus. A sensitivity analysis ( IX) confirmed the model's accuracy over 90-100% thresholds, validating this heuristic. These results confirm the resilience of our threshold, although further systematic evaluations across

datasets and model architectures are needed. We plan to study adaptive or probabilistic consensus procedures in multi-model ensembles with more than two LLMs.

- Interpretability: While our current focus is on accuracy, future work could incorporate interpretability mechanisms like attention maps or saliency overlays to provide deeper insights into the models' decision-making processes, further enhancing clinical trust.

- Computational Cost and Disagreement: A detailed analysis of the computational overhead associated with running multiple models in parallel is needed to assess practical feasibility. Future work will also explore automated strategies for resolving disagreements, rather than simply flagging them for review. Also, we plan to test on real EHR-derived reports in future work.

TABLE IX: Sensitivity analysis of consensus accuracy across BERTScore thresholds.

| Threshold | Consensus Count | Accuracy (%) |
| --- | --- | --- |
| 0.90 | 50 | 91.3 |
| 0.925 | 50 | 91.3 |
| 0.95 | 47 | 91.3 |
| 0.975 | 44 | 90.9 |
| 1.00 | 41 | 90.2 |

## VI. CONCLUSION

We present a model-agnostic consensus framework that coordinates multiple large language models using structured prompting and semantic alignment to enhance diagnostic reliability in medical image interpretation. By integrating synthetic clinical context and leveraging inter-model agreement, our approach improves diagnostic confidence without requiring model retraining. This work highlights that consensus between independently trained LLMs can serve as a strong proxy for trustworthiness. The system's ability to flag uncertain or conflicting cases for human review introduces a valuable safeguard for clinical deployment. Future work will extend this framework to real-world clinical notes, additional imaging modalities, and domain-specific open-source models to further enhance robustness and generalizability.

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
