# OpenReview forum: "Fusion-Augmented Large Language Models: Boosting Diagnostic Trustworthiness via Model Consensus"
_IEEE.org/EMBS/BHI/2025/Conference — BHI 2025_

### Official Review · Reviewer_1TuY · 2025-06-27
**Fusion-Augmented Large Language Models: Boosting Diagnostic Trustworthiness via Model Consensus**

**Confidence:** 4
**Clarity Of Writing:** good
**Clinical Significance:** good
**Methodological Novelty:** good
**Overall Rating:** 6
**Final Rating:** 7

**Experiments And Results:**

fair

**Questions For The Authors:**

What is the rationale behind the 95% similarity threshold for consensus? Did you evaluate other thresholds, and how sensitive are your results to this parameter?

Could you elaborate on how disagreement cases (i.e., when consensus is not reached) were handled? This is important for real-world deployment.

Do you plan to test this fusion method on real multimodal data (e.g., actual clinical notes + images) rather than synthetic ones? That would improve generalizability.

Have you considered including interpretability mechanisms (e.g., attention maps or saliency overlays) to improve clinical trust beyond accuracy metrics?

Can you provide more insight into model disagreement patterns? For example, do Claude and ChatGPT disagree more often on certain pathologies or subtleties?

**Strengths:**

Addresses a critical challenge in AI healthcare: trustworthiness and diagnostic variability among large models.

The consensus-based fusion approach is intuitive, computationally lightweight, and effective.

Strong empirical results: the multimodal fusion achieved a notable 91.3% accuracy, demonstrating meaningful performance gains.

Realistic synthetic notes based on the MIMIC-CXR template bring the evaluation closer to practical clinical use cases.

Evaluation on a radiologist-annotated dataset (CheXpert) adds clinical relevance and rigor to the benchmarking.

The approach is reproducible and model-agnostic, which enhances its potential for generalization.

**Summary Of The Paper:**

This paper introduces a fusion-based framework for improving diagnostic accuracy and trustworthiness in chest X-ray interpretation using large language models (LLMs). Specifically, the authors compare individual and combined performance of ChatGPT and Claude on a subset of radiologist-annotated CheXpert data using image-only prompts and a synthetic clinical text–augmented multimodal setup. A similarity-based consensus mechanism (95% output similarity) was applied to generate final predictions. Unimodal consensus improved diagnostic accuracy from 62.8% (ChatGPT) and 76.9% (Claude) to 77.6%. In the multimodal setting with synthetic notes, consensus accuracy reached 91.3%, outperforming individual models. The study demonstrates the potential of output-level consensus and multimodal inputs to boost trust and performance in AI-driven clinical diagnosis.

**Weaknesses:**

The study lacks detailed qualitative or error analysis (e.g., what types of findings were commonly misclassified by each model or the fusion method).

The output similarity threshold (95%) used for consensus is somewhat arbitrary—no justification or sensitivity analysis is presented.

Although synthetic notes are modeled after MIMIC-CXR, the impact of synthetic vs. real clinical text remains unclear.

The evaluation set sizes (234 for unimodal, 50 for multimodal) are relatively small given the scale of the CheXpert dataset; more extensive testing would boost confidence.

No discussion is provided on how disagreements between models are handled when consensus is not met (e.g., fallback strategies or uncertainty modeling).

---

### Official Review · Reviewer_bx1g · 2025-07-06
**Review of “Fusion-Augmented Large Language Models: Boosting Diagnostic Trustworthiness via Model Consensus”**

**Confidence:** 4
**Clarity Of Writing:** good
**Clinical Significance:** great
**Methodological Novelty:** good
**Overall Rating:** 5
**Final Rating:** 6

**Experiments And Results:**

good

**Questions For The Authors:**

See the weakness part, provide some explanation, and improvement.

**Strengths:**

The authors systematically conducted both unimodal (image-only) and multimodal (image plus synthetic clinical notes) diagnostic experiments. The results demonstrate that consensus among multiple models can significantly improve diagnostic accuracy, offering a notable advantage over single-model predictions. The proposed approach requires no fine-tuning of the underlying LLMs, exhibits strong model compatibility, and is readily applicable to real-world radiology AI workflows. The strengths of this work lie in its focus on model reliability and interpretability, its comprehensive validation on a widely recognized public dataset, and the introduction of a semantic similarity-based consensus mechanism that greatly enhances the interpretability of the outputs. Experimental results further show that the consensus framework is particularly effective in multimodal settings, where the integration of image and clinical context leads to improved diagnostic accuracy and robustness.

**Summary Of The Paper:**

This study proposes a framework that leverages two state-of-the-art large language models, ChatGPT and Claude, for multi-model fusion to enhance the reliability of chest X-ray interpretation on the CheXpert dataset. Overall, this article is well-written and interesting.

**Weaknesses:**

W1: The multimodal analysis was conducted on only 50 cases with synthetic clinical notes, which does not fully capture the diversity and complexity of real clinical narratives.
W2: The study’s focus on just two proprietary LLMs limits the generalizability of the findings to broader settings. The selection of the consensus threshold is somewhat empirical and lacks systematic justification.
W3: Furthermore, there is an insufficient quantitative analysis of the computational cost associated with consensus fusion and a lack of in-depth error analysis on disagreement cases, both of which are important for guiding future clinical adoption.
W4: As these prompts are based on GPT-4 and other advanced LLMs that are state-of-the-art at the time of writing, they may quickly become outdated due to the rapid evolution of model architectures. What does the author think about this?

---

### Official Review · Reviewer_9ZkZ · 2025-07-15
**review for Fusion-Augmented Large Language Models: Boosting Diagnostic Trustworthiness via Model Consensus**

**Confidence:** 4
**Clarity Of Writing:** great
**Clinical Significance:** good
**Methodological Novelty:** good
**Overall Rating:** 6
**Final Rating:** 6

**Experiments And Results:**

fair

**Questions For The Authors:**

**Sample selection criterion:** How the 234 curated X rays picked from the pool of 224,316 radiographs? Same with the subset of 50.

**Comparative analysis:**  Can you provide unimodal results for these same 50 cases to enable fair comparison? Does the higher correctness seen from GPT model from the bias on the 50 or the multimodal effect?

**Consensus metric:** Why use BERTScore for binary outputs instead of simpler exact match? Have you tested alternative agreement metrics?

**Model specification:** Is both GPT-4 and GPT-4o-mini used?

Answering the above question may help address major concerns raised in the weakness section.

**Strengths:**

The paper introduces an LLM-based consensus framework that attempts to mimic practical clinical workflows where multiple radiologists review cases to reduce diagnostic errors. The simplicity of the framework architecture and no requirement of model finetuning and retraining make it fast implementable for real-life uses. In addition, the approach is inherently model-agnostic, making it adaptable to the rapidly evolving LLM landscape and scalable beyond the two-model setup tested here. The overall writing is clear and the methodology provides sufficient details to understand.

**Summary Of The Paper:**

The paper recognizes the need for more trustworthy diagnostic outcomes from radiology data using LLMs. To address this, it proposes a consensus-based framework that combines outputs from ChatGPT and Claude for chest X-ray interpretation on the CheXpert dataset. The approach uses semantic similarity (BERTScore with 95% threshold) to identify cases where both models agree, treating agreement as a signal of reliability. The authors evaluate both unimodal (image-only) and multimodal (image + synthetic clinical notes) conditions on 234 and a subset of 50 cases respectively. Multimodal consensus accuracy reaches 91.3%, while unimodal consensus accuracy reaches 77.6%, both higher than results from individual models. The paper suggests that the model agreement can serve as a proxy for trustworthiness and such consensus-based fusion improves diagnostic reliability without requiring model finetuning and retraining.

**Weaknesses:**

**Missing sample selection criterion:** The paper lacks reasoning on the selection criteria or methodology of the 234 and 50 cases from the pool of 224,316 images.

**Unfair comparative analysis:** the multimodal result used a subset of 50 cases from the 234, but no clarification or analysis on how the unimodal correctness look like on these 50.

**Unjustified consensus metric:** BERTScore is unnecessarily complex for comparing binary outputs (0/1 for 14 categories). Simple exact match would be more appropriate and interpretable than a text similarity metric.

**Model specification concerns:** The paper mentions "ChatGPT," "GPT-4," and "GPT-4o-mini" in different sections. If multiple models are used, please clarify, otherwise, use a consistent model name. And since it is using API access, it may not be called “ChatGPT”, please consider use GPT or GPT-4o (consistent with the model used)

**Minor wording/clarity issues:**
* Section 2A LLMs in Medical Decision-Making, overstates this sentence “they often hallucinate findings or miss subtle abnormalities, especially with limited context”. The cited paper did not say that LLMs “often” hallucinate. Please consider rephrasing and rewording the sentence.
* the last paragraph of Section 2A LLMs in Medical Decision-Making is a bit disconnect from the related worked mentioned, suggest more specific sentence like, “collaborative advisor in radiology? In clinical decision making?” to show that your attempt is to fill the gap or address the problem current work has not done yet.
* In Section 2B Ensemble Methods in AI, “In medical imaging, ensembles are common in benchmarks such as CheXpert [8]”:  Consider cite more ensemble approaches not just cite CheXpert

---

### Official Review · Reviewer_nCVy · 2025-07-18
**Review of "Fusion-Augmented Large Language Models: Boosting Diagnostic Trustworthiness via Model Consensus"**

**Confidence:** 2
**Clarity Of Writing:** excellent
**Clinical Significance:** good
**Methodological Novelty:** fair
**Overall Rating:** 6

**Experiments And Results:**

great

**Questions For The Authors:**

1. Would it be possible to extend this method to integrate more than two LLMs? What challenges might arise in such a scenario, and what potential solutions could be considered to address these challenges?

**Strengths:**

1. The motivation, problem setting, contributions, methods, and results are all clearly presented in a well-structured manner.

2. This work addresses an important topic by utilizing large language models (LLMs) for medical image processing.

3. The evaluation metrics are well presented.

4. The results are detailed.

**Summary Of The Paper:**

This work proposes a fusion method that combines ChatGPT and Claude to enhance the reliability of chest X-ray interpretation on the CheXpert dataset. It addresses the limitations of current large language models (LLMs) in medical image processing and demonstrates improved accuracy in X-ray pattern recognition tasks during evaluation.

The motivation and problem setting are clearly defined and well-presented, and the overall manuscript is well-structured.

**Weaknesses:**

1. The novelty of the fusion method is fair.